# Trimester-specific phthalate exposures in pregnancy are associated with circulating metabolites in children

Jaclyn M. Goodrich[1], Lu Tang[2], Yanelli R. Carmona[3], Jennifer L. Meijer[3,4], Wei Perng[3,5], Deborah J. Watkins[1], John D. Meeker[1], Adriana Mercado-García[6], Alejandra Cantoral[7], Peter X. Song[2], Martha M. Téllez-Rojo[6], Karen E. Peterson[1,3]*

1 Department of Environmental Health Sciences, University of Michigan School of Public Health, Ann Arbor, MI, United States of America, 2 Department of Biostatistics, University of Pittsburgh, Pittsburgh, PA, United States of America, 3 Department of Nutritional Sciences, University of Michigan School of Public Health, Ann Arbor, MI, United States of America, 4 Department of Medicine, Dartmouth-Hitchcock Medical Center, Lebanon, NH, United States of America, 5 Department of Epidemiology, Colorado School of Public Health, Aurora, CO, United States of America, 6 Center for Research on Nutrition and Health, National Institute of Public Health, Cuernavaca, Morelos, México, 7 Department of Health, Universidad Iberoamericana, Mexico City, Mexico

* karenep@umich.edu

**Data Availability Statement:** All data files are available from the University of Michigan Library's

# Abstract

## Background

Prenatal phthalates exposures have been related to adiposity in peripuberty in a sex-specific fashion. Untargeted metabolomics analysis to assess circulating metabolites offers the potential to characterize biochemical pathways by which early life exposures influence the development of cardiometabolic risk during childhood and adolescence, prior to becoming evident in clinical markers.

## Methods

Among mother-child dyads from the Early Life Exposure in Mexico to ENvironmental Toxicants (ELEMENT) birth cohort, we measured 9 phthalate metabolites and bisphenol A in maternal spot urine samples obtained during each trimester of pregnancy, corrected for urinary specific gravity and natural log-transformed. In 110 boys and 124 girls aged 8–14 years, we used a mass-spectrometry based untargeted metabolomics platform to measure fasting serum metabolites, yielding 572 annotated metabolites. We estimated the associations between trimester-specific urinary toxicants and each serum metabolite, among all children or stratified by sex and adjusting for child age, BMI z-score, and pubertal onset. We accounted for multiple comparisons using a 10% false discovery rate (q<0.1).

## Results

Associations between exposures and metabolites were observed among all children and in sex-stratified analyses (q<0.1). First trimester MEP, MiBP, and MCPP were associated with decreased 2-deoxy-D-glucose among all children. Among girls, third trimester

Deep Blue Data repository (location: https://doi.org/
10.7302/pehh-r785).

**Funding:** This study was funded by grants from the
U.S. Environmental Protection Agency (US EPA,
https://www.epa.gov/), grant numbers RD834800
(KEP) and RD83543601 (KEP), and from the
National Institute for Environmental Health
Sciences (NIEHS, https://www.niehs.nih.gov/) P20
ES018171 (KEP), P01 ES02284401 (KEP), R01
ES007821 (KEP), R01 ES014930 (MMTR), R01
ES013744 (MMTR), and P30 ES017885 (KEP,
JMG). The Michigan Regional Comprehensive
Metabolomics Resource Core, funded by grants
from the National Institute of Diabetes and
Digestive and Kidney Diseases (https://niddk.nih.
gov) completed the metabolomics analysis (grant
numbers P30 DK089503 and U24 DK097153).
This study was also supported and partially funded
by the National Institute of Public Health/Ministry of
Health of Mexico (https://www.insp.mx/insp-
overview.html; MMTR). There was no additional
external funding received for this study. The
contents of this publication are solely the
responsibility of the grantee and do not necessarily
represent the official views of the funding agency.
The funders had no role in study design, data
collection and analysis, decision to publish, or
preparation of the manuscript.

**Competing interests:** The authors have declared
that no competing interests exist.

concentrations of MECPP, MEHHP, MEHP, and MCPP were associated with 15, 13, 1, and
10 metabolites, respectively, including decreased choline and increased acylcarnitines and
saturated FAs (FA). Among boys, third trimester MIBP was positively associated with 9 fea-
tures including long chain saturated FAs, and second trimester MBzP was inversely associ-
ated with thyroxine.

## Conclusions

Metabolomics biomarkers may reflect sex- and exposure timing-specific responses to pre-
natal phthalate exposures manifesting in childhood that may not be detected using standard
clinical markers of cardiometabolic risk.

## Introduction

Phthalate esters and bisphenol A (BPA) are endocrine disrupting chemicals (EDCs) and plasti-
cizers found in a wide range of consumer products including but not limited to food packag-
ing, thermal paper, polycarbonate plastics, vinyl flooring, and personal care products [1–3].
These chemicals are detectable in samples from populations all around the world including in
pregnant women [4–8]. Exposure during pregnancy is particularly concerning as EDCs have
been implicated in the developmental origins of health and disease, especially as it relates to
risk for metabolic disorders. In epidemiological and rodent studies, prenatal phthalate or BPA
exposures have been associated with effects on adiposity, pubertal timing, hormone levels, and
metabolic biomarkers in childhood and adolescence [4, 9–16]. Relationships vary by toxicant,
sex, age of children assessed, and timing of exposure (e.g., trimester-specific). There is evi-
dence for gestation, and even certain periods of gestation, as critical windows of susceptibility
to the influence of EDCs on metabolic health. In the Early Life Exposure in Mexico to Environ-
mental Toxicants (ELEMENT) study, associations between prenatal exposures to phthalates or
BPA and adiposity and some metabolic outcomes have been reported in peripubertal children
[4, 11, 15–19]. Outcomes associated with prenatal phthalate exposures included adiposity [11],
BMI trajectories from 8 to 14 years of age [19], pubertal onset and reproductive hormone levels
[15, 16, 18], and fasting glucose and insulin-like growth factor 1 (IGF-1) [4]. While these stud-
ies link prenatal exposures to phthalates to adverse outcomes later in life, our understanding of
biological mechanisms leading to these long-term effects is limited.

The metabolome, encompassing all metabolites in a biological sample, provides insight on
the physiology of the system. Metabolomics has vast potential to improve risk assessment of
chemical exposures by augmenting our understanding of mechanisms of toxicity and disease,
and inter-individual susceptibility to toxicity. Ultimately, this knowledge would inform pre-
vention (i.e. reducing exposures) or intervention strategies to improve cardiometabolic health
[20]. Rodent and human studies suggest that effects from phthalates and BPA are detectable in
the metabolome [21–24], though longitudinal studies on prenatal exposures are limited. In
one study, adult rodents were exposed for four weeks to one of three doses of dibutylphthalate
(DBP) or one high dose of di(2-ethylhexyl)phthalate (DEHP). Dose-dependent (for DBP) and
sex-specific alterations (for DBP and DEHP) to the metabolome were reported with a greater
number of statistically significant findings in the males [21]. In a study using human-relevant
exposure levels, rats treated with diethyl phthalate (DEP) from gestation through adulthood
had 48 altered metabolites involved in pyruvate metabolism, sulfate conjugation, and other
pathways compared to controls [22]. In a cross-sectional study of Chinese adult males,

associations between biomarkers for DEHP and DEP exposure and the urinary metabolome were observed [23]. Taken together, these animal and human studies demonstrate the potential short-term impact of phthalate exposure on multiple classes of lipids, peroxisome proliferation, and amino acid metabolism. However, whether prenatal exposure to these toxicants results in altered metabolism years later in childhood or adolescence remains unknown.

Utilizing rich data from the ELEMENT cohort study, we aim to address knowledge gaps regarding the metabolite profiles in children associated with exposure to phthalates or BPA assessed in maternal urine from first, second, and third trimesters of pregnancy (T1, T2, T3). The overall goal is to elucidate altered biochemical pathways in childhood that may serve as the link between exposures and adverse cardiometabolic outcomes. We hypothesize that prenatal exposures will be associated with metabolite profiles relevant to metabolic risk among children. We hypothesize that associations will depend on the sex of the child and the trimester of the exposure. We test these hypotheses using untargeted metabolomics to profile the serum metabolome of 234 children (124 girls and 110 boys ages 8–14 years) from the ELEMENT study.

## Results

### Characteristics of the study population

At enrollment during pregnancy, mothers were 27 years old on average, had completed a median of 11.3 years of education and 71% were married or living with a partner (Table 1). The urine concentration of BPA and phthalates during pregnancy (pre- and post-imputation) are presented in S1 Table. The median age of the 234 children was 9.9 years, and 53% were girls. Median age was the same for both boys and girls (9.9 years). Pubertal onset (Tanner stage >1) was documented in 39% of the children (32% of girls and 46% of boys). The prevalence of obesity and overweight were 18.2% and 22.4%, respectively, according to the WHO criteria [25].

**Table 1. Characteristics of the study population (n = 234).**

| Variable | | N (%) | Mean (SD) or Median (IQR)* |
|---|---|---|---|
| *Maternal characteristics during pregnancy*: | | | |
| Maternal age (years) | | | 27.0 (5.7) |
| Maternal Education (years completed) | | | 11.5 (9, 12) |
| Marital Status | Married | 167 (71.4%) | |
| | Other | 67 (28.6%) | |
| Cohort of ELEMENT | 2nd | 58 (24.8%) | |
| | 3rd | 176 (75.2%) | |
| *Child characteristics*: | | | |
| Sex | Boys | 110 (47.0%) | |
| | Girls | 124 (53.0%) | |
| *At time of sample collection for metabolomics*: | | | |
| Child's age (years) | | | 9.9 (8.8, 11) |
| BMI Z-score | | | 0.862 (1.243) |
| Pubertal onset | Yes | 91 (38.9%) | |
| | No | 143 (61.1%) | |

*Mean and SD reported for normally distributed variables. Median, Q1 and Q3 are reported otherwise (for maternal education and child's age).

IQR = interquartile range; SD = standard deviation

## Toxicants and metabolites: All children

Linear models were run to examine associations between nine trimester-specific urinary phthalate concentrations or BPA from T1, T2, and T3 and 572 known metabolites measured in serum of 234 children. S2 Table summarizes the number of associations between exposure and metabolites for models with all children and sex-stratified models that are statistically significant (q<0.1). Among all children, four phthalates were significantly associated with metabolites (Table 2). MBP, MEP and MCPP in T1 were all associated with decreased 2-deoxy-D-glucose (beta coefficient ± standard error = -0.31±0.08). MCPP was also associated with increased phosphoserine 41:7 (β = 0.27±0.07). No toxicants assessed in T2 were significantly associated with metabolites. In T3, MBP was associated with increased homoserine (β = 0.26 ±0.07) and MIBP with increased medium and long chain saturated FAs (β = 0.25 to 0.26). There were no significant associations between BPA or other phthalates and metabolites (all q>0.1).

## Toxicants and metabolites: Girls

Statistically significant relationships between prenatal phthalate concentrations (specifically, MCPP and biomarkers of DEHP exposure during T3) and metabolites were observed among 124 girls (Table 3). MCPP was positively associated with 10 metabolites including acylcarnitines and FAs from the keto pathway (i.e. 7-oxo-11E-tetradecenoic acid). Biomarkers of DEHP exposure (MECPP, MEHHP, and MEHP) displayed significant associations with 15, 13, and 1 known metabolites, respectively. Diacylglycerol 34:5 was inversely associated with all three (βs = -0.36 to -0.27). Seven other metabolites were associated with MECPP and MEHHP: decreased choline and sn-glycero-3-phosphocholine (βs = -0.35 to -0.28); increased phenylalanine dipeptide, phosphatidylinositol 38:1, dicarboxylate FA 14:0, riboflavin, and testosterone.

To identify clusters of correlated metabolites that might be influenced by phthalate exposure, we assessed the relationship between exposure-associated metabolites with each other.

**Table 2. Significant associations (q<0.1) between maternal trimester-specific phthalates and metabolites among 234 children.**

| Exposure | Trimester of Exposure Measure | Metabolite Name** | Model Estimate* (SE) | p-value | q-value | Mass | Retention Time | Ionization Mode | Sub Pathway |
|---|---|---|---|---|---|---|---|---|---|
| MEP | T1 | 2-deoxy-D-glucose | -0.305 (0.08) | 1.80E-04 | 0.099 | 164.0684 | 0.7048 | N | Fructose, Mannose and Galactose Metabolism |
| MBP | T1 | 2-deoxy-D-glucose | -0.311 (0.08) | 1.29E-04 | 0.074 | 164.0684 | 0.7048 | N | Fructose, Mannose and Galactose Metabolism |
| | T3 | homoserine | 0.256 (0.065) | 1.04E-04 | 0.059 | 119.0581 | 0.6843 | P | Glycine, Serine and Threonine Metabolism |
| MIBP | T3 | FA 14:0 | 0.248 (0.067) | 2.46E-04 | 0.051 | 228.2091 | 22.4071 | N | Long Chain Fatty Acid |
| | | FA 15:0 | 0.249 (0.068) | 3.02E-04 | 0.051 | 242.2245 | 22.6196 | N | Long Chain Fatty Acid |
| | | FA 12:0 | 0.263 (0.067) | 1.26E-04 | 0.051 | 200.1775 | 21.7155 | N | Medium Chain Fatty Acid |
| MCPP | T1 | phosphoserine 41:7 | 0.271 (0.067) | 7.20E-05 | 0.041 | 847.5315 | 24.3775 | P | Glycerophosphoserines |
| | | 2-deoxy-D-glucose | -0.307 (0.079) | 1.44E-04 | 0.041 | 164.0684 | 0.7048 | N | Fructose, Mannose and Galactose Metabolism |

*Estimate is for the exposure (first ln-transformed and standardized to mean 0, variance 1) in a model of the metabolite, adjusting for age, BMI z-score, pubertal onset (yes/no), and sex.

**FA = fatty acid

**Table 3. Significant associations (q<0.1) between maternal phthalate exposures during third trimester and metabolites among 124 girls.**

| Exposure | Metabolite Name** | Model Estimate*** (SE) | p-value | q-value | Mass | Retention Time | Ionization Mode | Sub Pathway |
|---|---|---|---|---|---|---|---|---|
| MCPP | AC 8:0 (OH) | 0.357 (0.092) | 1.64E-04 | 0.050 | 303.2045 | 7.9590 | P | Fatty Acid Metabolism(Acyl Carnitine), hydroxy |
| | AC 6:0 (OH) | 0.335 (0.089) | 2.66E-04 | 0.050 | 275.1739 | 3.5536 | P | Fatty Acid Metabolism(Acyl Carnitine), hydroxy |
| | AC 10:0 (OH) | 0.329 (0.091) | 4.42E-04 | 0.050 | 331.2357 | 12.4937 | P | Fatty Acid Metabolism(Acyl Carnitine), hydroxy |
| | Keto 14:0 | 0.325 (0.09) | 4.60E-04 | 0.050 | 242.1884 | 20.7967 | N | Fatty Acid, Keto |
| | AC 14:1 | 0.326 (0.091) | 4.64E-04 | 0.050 | 369.2874 | 18.9595 | P | Fatty Acid Metabolism(Acyl Carnitine) |
| | FA 12:0 (OH) | 0.333 (0.093) | 5.20E-04 | 0.050 | 216.1724 | 19.3867 | N | Fatty acid,hydroxy |
| | O-Acetylcarnitine | 0.303 (0.088) | 7.57E-04 | 0.062 | 203.116 | 0.9144 | P | Fatty Acid Metabolism; BCAA Metabolism |
| | AC 4:0 (OH) | 0.302 (0.089) | 9.33E-04 | 0.064 | 247.1422 | 1.1458 | P | Fatty Acid Metabolism(Acyl Carnitine), hydroxy |
| | FA 10:0 (OH) | 0.299 (0.089) | 1.01E-03 | 0.064 | 188.1411 | 16.3034 | N | Fatty acid,hydroxy |
| | 7-oxo-11E-tetradecenoic acid | 0.291 (0.09) | 1.51E-03 | 0.086 | 240.1727 | 19.6346 | N | Fatty Acid, Keto |
| MECPP | Sn-glycero-3-phosphocholine | -0.34 (0.083) | 7.88E-05 | 0.036 | 257.1031 | 0.6142 | P | Glycerophosphocholines |
| | dipeptide (phe phe) | 0.383 (0.099) | 1.78E-04 | 0.041 | 312.1484 | 8.4177 | P | Dipeptide |
| | PI 38:1 | 0.297 (0.086) | 7.76E-04 | 0.071 | 752.4347 | 22.0423 | P | Phosphatidylinositol |
| | FA 12:0 (OH) | 0.294 (0.088) | 1.19E-03 | 0.071 | 216.1724 | 19.3867 | N | Fatty acid,hydroxy |
| | choline | -0.278 (0.084) | 1.22E-03 | 0.071 | 104.1083 | 0.6329 | P | Phospholipid Metabolism |
| | Dicarboxylic FA 14:0 (OH) | 0.318 (0.096) | 1.24E-03 | 0.071 | 274.1783 | 15.6881 | P | Fatty Acid, Dicarboxylate, hydroxy |
| | DG 34:5 | -0.276 (0.084) | 1.36E-03 | 0.071 | 586.4577 | 24.8626 | P | Diacylglycerol |
| | Keto 14:0 | 0.28 (0.086) | 1.46E-03 | 0.071 | 242.1884 | 20.7967 | N | Fatty Acid, Keto |
| | riboflavin | 0.3 (0.092) | 1.50E-03 | 0.071 | 376.1322 | 14.1228 | N | Riboflavin Metabolism |
| | testosterone | 0.293 (0.09) | 1.54E-03 | 0.071 | 288.1938 | 17.0763 | P | Steroid |
| | PI 36:1 | 0.285 (0.09) | 1.88E-03 | 0.079 | 724.4045 | 21.8264 | P | Phosphatidylinositol |
| | AC 8:0 (OH) | 0.282 (0.09) | 2.16E-03 | 0.083 | 303.2045 | 7.9590 | P | Fatty Acid Metabolism(Acyl Carnitine), hydroxy |
| | L-histidine | -0.26 (0.086) | 2.91E-03 | 0.096 | 155.0693 | 0.6378 | N | Histidine Metabolism |
| | tripeptide (gly pro val) | 0.274 (0.09) | 3.04E-03 | 0.096 | 271.1541 | 2.8976 | P | Tripeptide |
| | Dicarboxylic FA 13:0 (OH) | 0.274 (0.091) | 3.13E-03 | 0.096 | 260.1627 | 14.1495 | P | Fatty Acid, Dicarboxylate, hydroxy |

(*Continued*)

**Table 3.** (Continued)

| Exposure | Metabolite Name** | Model Estimate*** (SE) | p-value | q-value | Mass | Retention Time | Ionization Mode | Sub Pathway |
|---|---|---|---|---|---|---|---|---|
| MEHHP | Sn-glycero-3-phosphocholine | -0.352 (0.087) | 8.96E-05 | 0.039 | 257.1031 | 0.6142 | P | Glycerophosphocholines |
| | choline | -0.336 (0.086) | 1.64E-04 | 0.039 | 104.1083 | 0.6329 | P | Phospholipid Metabolism |
| | riboflavin | 0.335 (0.096) | 6.77E-04 | 0.076 | 376.1322 | 14.1228 | N | Riboflavin Metabolism |
| | Dicarboxylic FA 14:0 (OH) | 0.342 (0.1) | 8.64E-04 | 0.076 | 274.1783 | 15.6881 | P | Fatty Acid, Dicarboxylate, hydroxy |
| | DG 34:5 | -0.299 (0.087) | 8.73E-04 | 0.076 | 586.4577 | 24.8626 | P | Diacylglycerol |
| | 1,2-dipalmitoyl-sn-glycerol | -0.3 (0.09) | 1.22E-03 | 0.076 | 568.5067 | 26.2195 | P | Diacylglycerol |
| | dipeptide (phe phe) | 0.347 (0.105) | 1.27E-03 | 0.076 | 312.1484 | 8.4177 | P | Dipeptide |
| | PA 25:3 | 0.285 (0.088) | 1.58E-03 | 0.076 | 530.3468 | 21.4930 | N | Glycerophosphates |
| | testosterone | 0.305 (0.094) | 1.59E-03 | 0.076 | 288.1938 | 17.0763 | P | Steroid |
| | PI 38:1 | 0.292 (0.09) | 1.60E-03 | 0.076 | 752.4347 | 22.0423 | P | Phosphatidylinositol |
| | Dicarboxylic FA 15:0 (OH) | 0.29 (0.091) | 1.76E-03 | 0.077 | 288.1942 | 17.0413 | N | Fatty Acid, Dicarboxylate, hydroxy |
| | dipeptide | 0.349 (0.111) | 2.01E-03 | 0.079 | 278.1628 | 7.8247 | P | Dipeptide |
| | tripeptide (tyr trp leu) | 0.299 (0.095) | 2.14E-03 | 0.079 | 457.2177 | 4.0797 | P | Tripeptide |
| MEHP | DG 34:5 | -0.363 (0.094) | 1.91E-04 | 0.092 | 586.4577 | 24.8626 | P | Diacylglycerol |
| ΣDEHP* | DG 34:5 | -0.311 (0.086) | 4.42E-04 | 0.095 | 586.4577 | 24.8626 | P | Diacylglycerol |
| | Sn-glycero-3-phosphocholine | -0.31 (0.087) | 5.28E-04 | 0.095 | 257.1031 | 0.6142 | P | Glycerophosphocholines |
| | FA 12:0 (OH) | 0.32 (0.091) | 6.08E-04 | 0.095 | 216.1724 | 19.3867 | N | Fatty acid,hydroxy |

*Molar sum of DEHP biomarkers: MEHP, MEHHP, MEOHP, MECPP

**FA = fatty acid; (OH) = hydroxy; AC = acyl carnitine; PI = phosphatidylinositol; PA = glycerophosphate; DG = diacylglycerol

***Estimate is for the exposure (first ln-transformed and standardized to mean 0, variance 1) in a model of the metabolite, adjusting for age, BMI z-score, and pubertal onset (yes/no). The only statistically significant associations were between T3 phthalates and metabolites which are shown here.

Figs 1, 2, S1 and S2 Figs display these clusters; we refer to them hereafter by their labels within the figures. There were 2 clusters of highly correlated metabolites associated with T3 MCPP exposure: 1) a FA metabolism cluster (acylcarnitines and other carnitines, decenoic acids and other 10 to 16 carbon FAs) that was positively associated with T3 MCPP (Cluster 1 in Fig 1) and 2) diacylglycerols (DGs) that were inversely associated with T3 MCPP. Correlation analysis of metabolites associated with the DEHP biomarkers MECPP (Fig 2) or MEHHP (S1 Fig) among girls were similar, showing 3 clusters. The largest cluster (Cluster 2 in the figures) included testosterone, riboflavin, dicarboxylic FAs, phosphatidylinositols (PIs) and phosphatidic acids (PAs) that were correlated with one another and positively associated with MECPP and/or MEHHP. The next cluster contained acylcarnitines and medium to long-chain FAs,

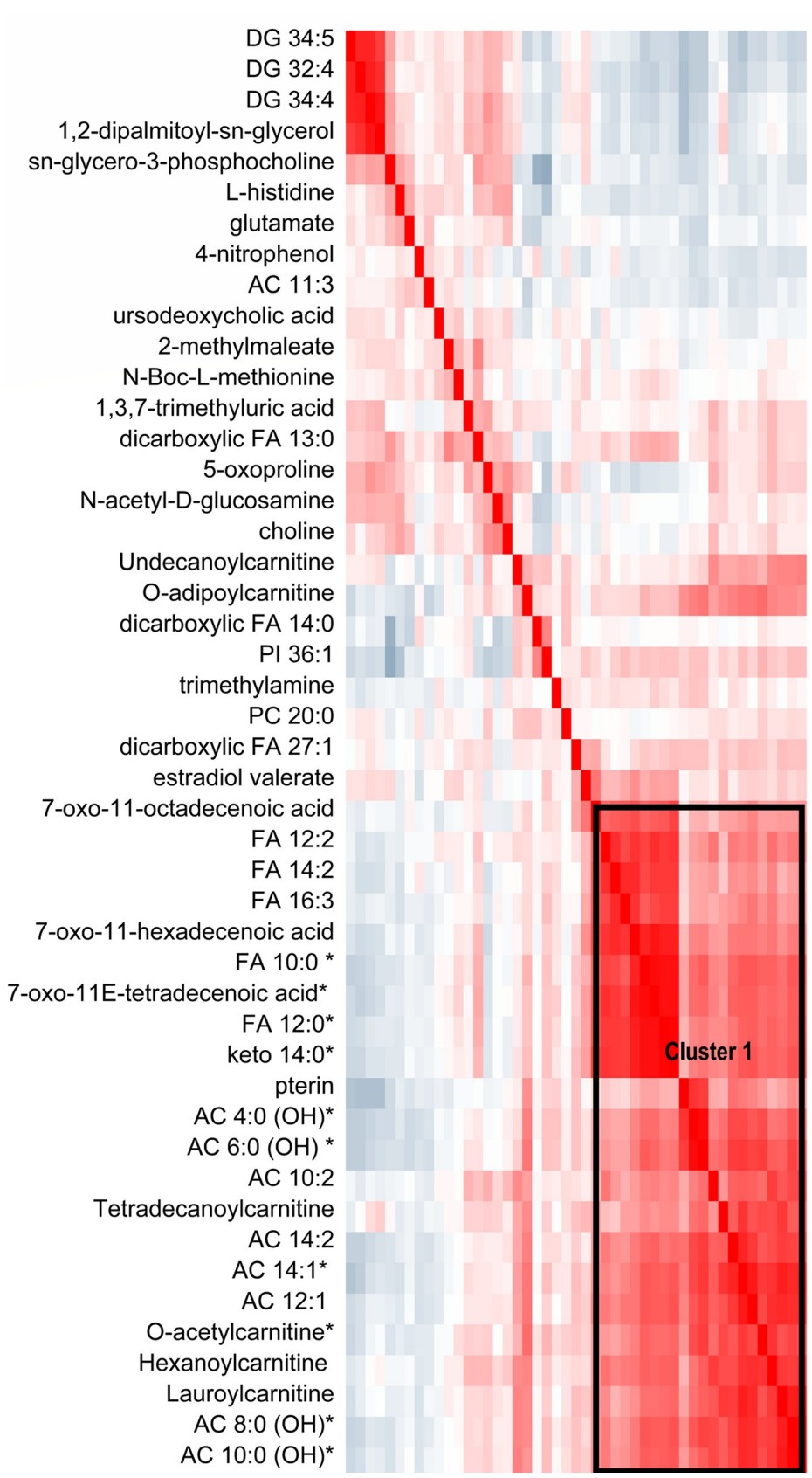

**Fig 1. Clustering of metabolites associated with third trimester maternal urinary mcpp among girls.** Metabolites that had p-values<0.05 for their association with T3 MCPP exposure among girls in childhood are included in the heatmap below; an *is next to the name of metabolites significant at q-value<0.1. The heatmap shows the Pearson correlation between these metabolites with each other, and metabolites are ordered by hierarchal clustering. The blue-gray color indicates an inverse correlation while red indicates a positive correlation. The Cluster 1 box indicates a group of metabolites that are strongly positively correlated with one another.

which were also positively associated with T3 exposure (Cluster 3). The last cluster includes choline, histidine, and diacylglycerols which were inversely associated with exposure (Cluster 4).

## Toxicants and metabolites: Boys

There were some statistically significant associations between metabolites and prenatal phthalate exposures among 110 boys (Table 4). In T2, MBP was positively associated with a glycero-phosphocholine/ethanoloamine ($\beta = 0.44\pm0.09$), and MBzP was inversely associated with thyroxine ($\beta = -0.34\pm0.09$) and a 16-chain dicarboxylic FA ($\beta = -0.35\pm0.09$). In T3, MIBP was associated with lower glucose ($\beta = -0.28\pm0.08$) as well as increased long and medium chain FAs and a 17-chain unsaturated dicarboxylic FA. Cluster analysis of the T3 MIBP results revealed 2 clusters of correlated metabolites (S2 Fig). The largest cluster consisted of long chain and very long chain FAs along with ceramides, DGs, phytanate and more (Cluster 5 in S2 Fig). T3 MIBP was positively associated with the metabolites in this cluster. The next set of metabolites (Cluster 6) included thyroxine, PCs, and PEs, which were inversely associated with T3 MIBP.

## Discussion

In this study, we observed associations between prenatal exposures to phthalates and the circulating metabolome in children ages 8 to 14 years that were in many cases trimester-, sex-, and exposure-specific. Phthalates are high production volume chemicals that humans are ubiquitously exposed to through consumer products and the built environment. The greatest number of statistically significant associations was observed in the girls-only analysis with T3 exposures including MCPP associated with acylcarnitines and biomarkers of DEHP exposure (MEHP, MEHHP and MECPP) associated with lipid-related metabolites. In the boys-only analysis, T2 or T3 MBP, MBzP, and MIBP were associated with long chain FAs and several other metabolites.

These findings inform our understanding of the biological processes impacted by phthalates that may underlie longer term cardiometabolic effects. Profiling the metabolome provides a composite measure of biological function and may enable identification of subclinical signs of toxicity. Most of the statistically significant associations were from sex-stratified analyses. However, several metabolites stood out in the analysis of all children together. Maternal urinary concentrations of MEP, MiBP, and MCPP in the first trimester were each associated with decreased 2-deoxy-D-glucose. This is a compound implicated in the metabolism of sugars (galactose, mannose, fructose) that is not broken down by glycolysis. Altered expression of glucose transporters, such as GLUT1, could explain the association between phthalates and 2-deoxy-D-glucose we observed. An *in vitro* study using a human pancreatic beta-cell line (1.1B4) found treatment of MEP led to increased expression of *GLUT1* [26]. Increased GLUT1 could lead to more transport of glucose and deoxy-D-glucose into cells (i.e. erythrocytes) resulting in lower levels in the serum as reflected here.

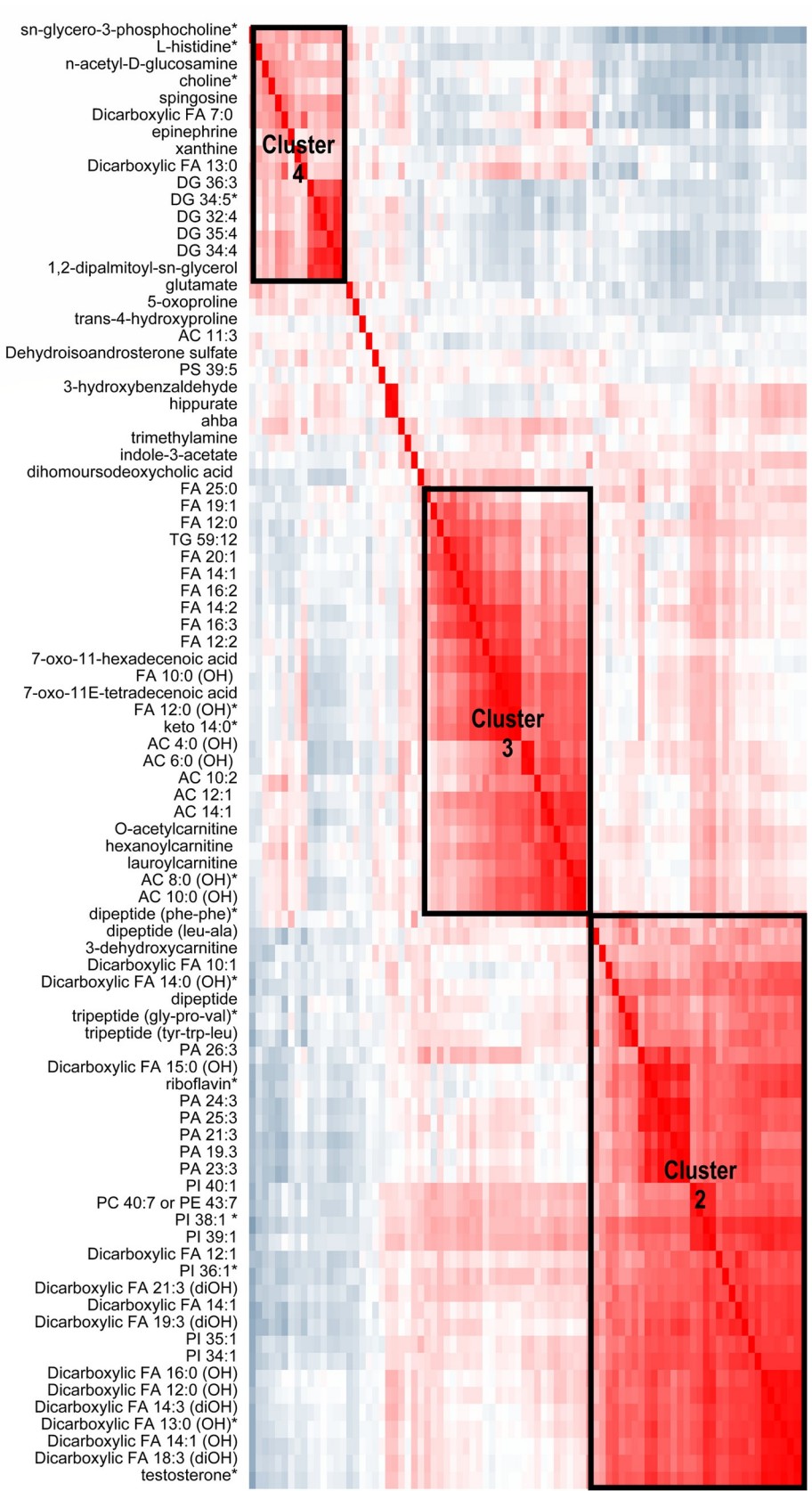

**Fig 2. Correlation among metabolites associated with maternal third trimester mecpp concentrations in peripubertal girls.** Metabolites that were associated with T3 MECPP among girls at an uncorrected p-value<0.05 are included in the heatmap below; an asterisk * is next to the name of metabolites significant at q-value<0.1. The heatmap shows the Pearson correlation between these metabolites with each other, and metabolites are ordered by hierarchal clustering. The blue-gray color indicates an inverse correlation while red indicates a positive correlation. The Cluster 2, 3, and 4 boxes denote groups of metabolites that are strongly positively correlated with one another.

Statistically significant associations (q<0.1) were observed in the girls-only analysis between T3 exposures (DEHP metabolites) with three major clusters of metabolites. A cluster including diacylglycerol 34:5, choline, sn-glycero-3-phosphocholine, and histidine were inversely associated with these phthalates (Cluster 4 in Fig 2 and S1 Fig). Diacylglycerols (DG) are lipids important for the structure of membranes, lipid metabolism and DG-dependent signaling. It has been suggested that alterations in the metabolism of DGs are related to metabolic diseases, including diabetes [27]. In our study, we found that ΣDEHP and individual DEHP metabolites were consistently and inversely associated with long-chain DG 34:5. This is consistent with a zebrafish study of embryonic administration of DEHP that reported reduced concentrations of saturated DGs, triglycerides, and other lipids [28]. DGs, choline, and sn-glycero-3-phosphocholine, which were inversely associated with DEHP biomarkers among girls, are connected to the choline metabolism pathway. Choline is important for cellular maintenance and growth, membrane synthesis, lipid transport, one-carbon metabolism, and

**Table 4. Significant associations (q<0.1) between maternal trimester-specific phthalates and metabolites among 110 boys.**

| Exposure | Metabolite Name* | Model Estimate* (SE) | p-value | q-value | Mass | Retention Time | Ionization Mode | Sub Pathway |
|---|---|---|---|---|---|---|---|---|
| MBP—T2 | PC 32:2 or PE 35:2 | 0.439 (0.092) | 6.61E-06 | 0.003 | 729.5338 | 23.8939 | P | Glycerophosphocholines or Glycerophosphoethanolamines |
| MIBP—T3 | FA 15:0 | 0.382 (0.092) | 6.86E-05 | 0.032 | 242.2245 | 22.6196 | N | Long Chain Fatty Acid |
| | FA 14:0 | 0.323 (0.087) | 3.05E-04 | 0.042 | 228.2091 | 22.4071 | N | Long Chain Fatty Acid |
| | glucose | -0.284 (0.076) | 3.12E-04 | 0.042 | 180.0636 | 0.6499 | N | Fructose, Mannose and Galactose Metabolism |
| | FA 12:0 | 0.318 (0.087) | 3.87E-04 | 0.042 | 200.1775 | 21.7155 | N | Medium Chain Fatty Acid |
| | PC 37:7 or PE 40:7 | -0.344 (0.095) | 4.52E-04 | 0.042 | 789.5631 | 24.9077 | P | Glycerophosphocholines or Glycerophosphoethanolamines |
| | Dicarboxylic FA 17:2 | 0.322 (0.092) | 6.92E-04 | 0.054 | 296.1984 | 22.4103 | N | Fatty Acid, Dicarboxylate |
| | FA 18:0 | 0.292 (0.087) | 1.05E-03 | 0.065 | 284.2719 | 23.1444 | N | Long Chain Fatty Acid |
| | FA 20:0 | 0.279 (0.083) | 1.12E-03 | 0.065 | 312.303 | 23.4789 | N | Long Chain Fatty Acid |
| | FA 16:0 | 0.285 (0.089) | 1.80E-03 | 0.093 | 256.2406 | 22.8039 | N | Long Chain Fatty Acid |
| MBzP—T2 | Dicarboxylic FA 16:0 | -0.349 (0.091) | 2.10E-04 | 0.063 | 286.2147 | 21.1076 | N | Fatty Acid, Dicarboxylate |
| | Thyroxine | -0.336 (0.094) | 5.75E-04 | 0.086 | 776.6865 | 16.1400 | P | Thyroxine Metabolism |

*PC = phosphocholine; PE = phosphoethanolamine; FA = fatty acid; T2 = second trimester; T3 = third trimester

**Estimate is for the exposure (first ln-transformed and standardized to mean 0, variance 1) in a model of the metabolite, adjusting for age, BMI z-score, and pubertal onset (yes/no).

neurotransmission [29]. The importance of decreased choline and related metabolites in Cluster 4 by prenatal DEHP exposure for long-term health, including cardiometabolic health and neurodevelopment, merits further consideration since studies using human and animal cell lines exposed to phthalates have shown deleterious effects of phthalates on the neuroendocrine system and neurodevelopment [30, 31].

For girls, DEHP exposure were also associated with increases in a cluster of metabolites including testosterone, phenylalanine dipeptide, PIs, PAs, dicarboxylic FAs, and riboflavin (Cluster 2 of Fig 2 and S1 Fig). Similar to our findings, in a study that analyzed urinary phthalates in the 3rd trimester and metabolomics among pregnant women, testosterone was found to be positively associated with the phthalates MCNP and MEP; yet metabolomics were measured in the mothers and not the children [32]. Phthalates are notorious for their disruption of reproductive hormones and outcomes [33]. The influence of prenatal exposure on peripubertal hormones is only recently coming to light. In this same ELEMENT study population, we reported that T3 MEHP was associated with early adrenarche (higher odds for Tanner stage>1 for pubic hair development), and prenatal MEP exposure was associated with greater serum testosterone among girls (ages 8–14 years). We also reported that T3 MBP, MiBP, ΣDEHP, and MCPP were associated with higher serum DHEA-S among girls [15]. Using the metabolomics data, T3 concentrations of the same phthalates were positively associated with DHEA-S, yet only the associations with DEHP biomarkers were near statistical significance. Collectively, data suggest that late gestational exposure to DEHP, and potentially other phthalates, increases androgens among peri-pubescent girls. These hormonal changes could be contributing to outcomes that were previously associated with T3 DEHP in ELEMENT girls: earlier onset of adrenarche and higher BMI trajectory by age 14 years [15, 19].

Third trimester biomarkers of DEHP were positively associated with additional metabolites in Cluster 2 among girls include PIs, PAs, and dicarboxylic acids. Of interest, T3 MECPP and MEHHP are positively associated with multiple saturated dicarboxylic FAs, which are formed during omega-oxidation of FAs or CoA esters. Omega oxidation increases due to excess fat intake (e.g. ketogenic conditions) or inadequate mitochondrial beta-oxidation [34]. In girls, T3 MCPP is associated with the third distinct cluster of metabolites including increased levels of acylcarnitines and medium to long chain hydroxy FAs (Cluster 3 in Fig 2 and S1 Fig). Acylcarnitines are produced during mitochondrial β-oxidation and are required for transport of fatty acyl-CoA esters across the mitochondrial membrane [35]. This may be one link to the known association between phthalate exposure and obesity risk given that beta oxidation is involved in numerous obesity-related physiological processes, including lipid metabolism, hepatic fat accrual, and glucose-insulin homeostasis [36, 37].

Among boys, third trimester MIBP was positively associated with six medium- and long-chain saturated FAs and a dicarboxylic FA (q<0.1 for each metabolite). A larger set of correlated metabolites related to lipid metabolism and synthesis including DGs, ceramides, lysophosphatidylethanolamines, lysoPCs, and medium to very-long chain FAs were positively associated with T3 MIBP among boys at a relaxed p<0.05 (S2 Fig, Cluster 5). The biological impact of increased medium and long-chain FAs around the time of puberty among boys is uncertain, and its relationship with adiposity may differ by age and diet. Previously in the ELEMENT cohort, the highest tertiles of MiBP, MBzP, MEHP, and MEHHP were associated with the lowest BMI by age 14 years [19]. Contrarily, positive associations between prenatal biomarkers of low- (MBP, MEP) and high-molecular weight (ΣDEHP, MBzP) phthalates with BMI and waist circumference (standardized to age and sex) were found in children (5–12 years of age) from a Mexican-American cohort. When stratifying by sex, MBP and MCPP were positively associated with BMI and waist circumference only in boys [38]. Among boys, we also observed an inverse association between T3 MIBP and glucose. This corroborates our

previous finding among pubertal boys from the same cohort using standard laboratory assessment of fasting glucose that showed lower glucose with greater prenatal DBP (MBP and MiBP) or MCPP exposure [17].

The biological mechanisms by which prenatal phthalate exposures alter metabolism and biological function through childhood and adolescence is not entirely known but may include peroxisome proliferator-activated receptor (PPAR) activation, disruption of sex steroid and thyroid hormone synthesis and signaling [39–41], and/or epigenetic programming (reviewed previously [42, 43]). Phthalates can activate PPARs—transcription factors that regulate genes involved in adipogenesis, lipid metabolism, and other biological pathways [44, 45]. Activation of these receptors during pregnancy could lead to aberrant transcription and epigenetic programming of PPAR-target genes which could influence gene regulation later in childhood [46]. Studies using candidate gene and epigenome-wide approaches have identified associations between prenatal phthalate exposure and DNA methylation at genes related to growth, endocrine function, metabolism, and inflammation [47, 48]. Since sex-specific associations in this study were almost exclusively with T3 exposures, it is possible this relationship with PPAR activation and its subsequent biological effects occur after sexual dimorphism during gestation. Although hormone fluctuations and the development of gonads extend through T3 in both sexes, subtle differences in timing could contribute to differences in effects of phthalates between the sexes [49].

This study used a novel discovery approach with untargeted metabolomics to elucidate pathways that may underlie the sex- and trimester-specific effects of phthalate exposures manifested later on in childhood. This study had limitations including sample size. Statistical power was greater in the girls-only analysis due to sample size. By nature, many metabolites are correlated with one another and the phthalate biomarkers are also correlated with one another within trimesters which may have led to overlap in some of the results. Urinary phthalates and BPA have short half-lives, and spot urine collection once during each trimester may not reflect typical exposure throughout that trimester. The one-time metabolomics measure is also a limitation as we are unable to make inferences about metabolic flux, and we did not include dietary information which could be a confounder. Not all metabolites can be assigned a unique identity (i.e. DG 34:5 could represent DG 18:2/16:3 or DG 18:3/16:2 as the double bond location cannot be determined with untargeted data). As such, targeted validation should be done in future studies for top metabolites of interest.

In summary, we report sex- and trimester-specific associations between biomarkers of prenatal phthalate exposures and metabolites in the serum of 8–14 year old children living in Mexico City. Altered metabolites include those involved in lipid metabolism, utilization, and transport and thyroid and reproductive hormones. Lipid metabolism and hormones have been associated with phthalates in animal and human studies of children or adults. This is one of the first studies to show this association with prenatal exposures, and findings build upon our understanding of how prenatal phthalate exposures may impact risk for cardiometabolic complications throughout life (Fig 3). Screening the metabolome instead of traditional lipid (i.e. cholesterol) or hormone panels enabled detection of potential biomarkers of toxicity from many pathways of interest. Altered metabolites may biological processes that lead to long-lasting health effects from prenatal exposures.

## Materials and methods

### Study sample

Participants for this study were 234 children age 8–14 years who were originally enrolled in two cohorts from the ELEMENT study. The ELEMENT project consists of 3 sequentially

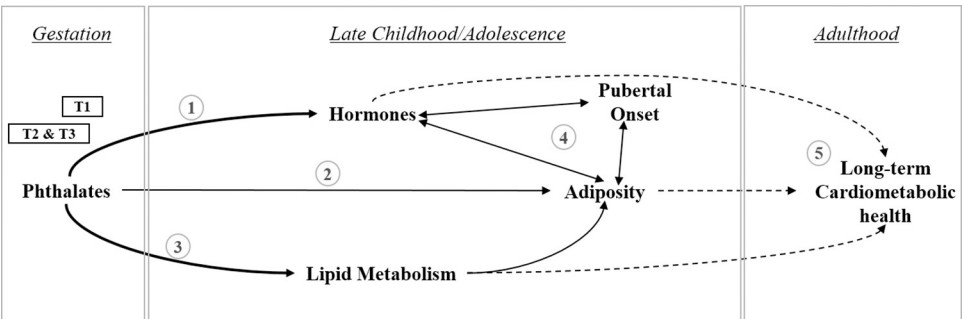

**Fig 3. Schematic of the relationship between gestational phthalate exposures, childhood outcomes, and long-term health.** Phthalate exposures during pregnancy have been associated with multiple biological responses in children that may underlie risk for cardiometabolic diseases later in adulthood. The graph below highlights connections with evidence from association studies including this study (thick arrows), other published studies (thin arrows), and hypothesized relationships (dashed-line arrows). T1 phthalate exposures, before sexual dimorphism, may influence all children; sex-specific associations with metabolomics were reported with exposures from T2 and T3. In summary, 1) previous ELEMENT studies and the current study identified associations between prenatal phthalate exposures and steroid hormone concentrations in peripuberty. 2) Previous studies reported sex-specific associations between prenatal phthalate exposures and adiposity in children/adolescents. 3) In this study, we identified relationships between prenatal phthalate exposures and metabolites including those involved in lipid metabolism and transport. 4) The pubertal transition is a time of rapid change when hormone levels, adiposity, pubertal onset, and metabolism all influence one another. 5) We hypothesize that these biological changes in childhood, including to the metabolome, are on the path leading to cardiometabolic risk from early-life phthalate exposures.

enrolled mother-child cohorts from three maternity hospitals in Mexico City. Families from Cohorts 2 and 3 that were followed-up through peripuberty and had archived samples from pregnancy for exposure assessment were included in this study [50–53]. Initially, women were recruited from 1997 to 2004 during pregnancy. Children were subsequently followed up throughout childhood. The subset of participants who had archived prenatal urine samples and were followed-up at a visit in 2011–2012 when the children were 8–14 years are included in this analysis (see [54] for further details). At the follow up visits, demographic and dietary data were collected via questionnaires, anthropometry measures were taken, and biospecimen (serum, spot urine samples, whole blood) were collected from the children as previously described [11].

Prior to participation, research staff explained study procedures to mothers and children. Mothers provided written consent upon enrollment in the study and at each follow-up visit, and children provided assent for the follow-up. The research protocol was approved by the Human Subjects Committee of the National Institute of Public Health of Mexico and the Internal Review Board at participating institutions including the University of Michigan.

## BPA and phthalate assessment in pregnancy samples

Total BPA and nine mono-ester metabolites of phthalate di-esters were analyzed in maternal spot urine samples from T1, T2, and T3 with isotope dilution-liquid chromatography- tandem mass spectrometry (ID-LC-MS/MS) at NSF International (Ann Arbor, MI, USA) as previously detailed [2, 16, 55]. The specific phthalate analytes were low molecular weight phthalates—monoethyl phthalate (MEP), mono-n-butyl phthalate (MBP), mono-isobutyl phthalate (MIBP)–and high molecular weight phthalates—mono(3-carboxypropyl) phthalate (MCPP), monobenzyl phthalate (MBzP), and four biomarkers of DEHP exposure: mono(2-ethylhexyl) phthalate (MEHP), mono(2-ethyl-5-hydroxyhexyl) phthalate (MEHHP), mono(2-ethyl-5-oxo-hexyl) phthalate (MEOHP), and mono(2-ethyl-5-carboxypentyl) phthalate (MECPP). Specific gravity of urine samples at each trimester was also measured using a digital refractometer.

## Metabolomics in child samples

The metabolome was assessed via an untargeted platform in fasting serum samples collected from ELEMENT children at the peri-pubertal study visit. Samples were immediately frozen following collection, and stored at -80˚C until analysis. At the Michigan Regional Comprehensive Metabolomics Resource Core, liquid chromatography mass spectrometry (LC-MS) was used to detect 9,303 features in positive or negative mode [56]. Briefly, metabolites were extracted using a solvent of 1:1:1 Methanol: Acetonitrile: Acetone with internal standards using 100 ml extraction solvent and 4 ml internal standard mixture. The 1290 Infinity Binary liquid chromatography system (Agilent Technologies, Inc., Santa Clara, CA) was used for chromatographic separation together with Waters Acquity UPLC HSS T3 1.8 μm 2.1 x 100 mm column in connection with a Water Acquity UPLC HSS T3 1.8 μm VanGuard pre-column. Samples were reconstituted with solvent (Methanol: Water, 2: 98). The total run time was 34 minutes with a flow rate of 0.45 mL/min and column temperature of 55 degrees C. The 6530 Accurate-Mass Q-TOF (Agilent Technologies, Inc., Santa Clara, CA) with a dual ASJ ESI ion source was used as the mass detector. Mass spectrometry was run in positive and negative ionization modes; the positive mode was run first, and six samples had missing negative mode data due to lack of remaining sample.

Raw data processing identified chromatographic peaks representing metabolite features using a modified version of existing commercial software (Agilent MassHunter Qualitative Analysis). After the removal of redundant compounds or those missing in >70% of samples, 3,758 features remained [57]. Metabolites were annotated by matching MS/MS fragmentation patterns, retention times, and ionization masses to metabolites within the laboratory's compound library, allowing for the annotation of 572 annotated compounds (denoted as 'known') that were used for analysis hereafter. Lipids are reported with the nomenclature as X:Y, where X is the length of the carbon chain and Y is the number of double bonds.

## Data analysis: Preprocessing and missing data imputation

Prior to statistical analysis, metabolomics data were normalized, adjusted for batch effects, and missing values were imputed within each model [37]. Within-batch peak intensities were adjusted for batch drift using LOESS regression; the extent of batch drift was determined by pooled samples used as quality control in each batch. Across-batch peak intensities were adjusted for drift by a multiplicative factor of the ratio between the median intensity of the batch-specific quality control samples and the median intensity of all quality control samples. Missing metabolite values were imputed using the K-nearest-neighbour algorithm (K = 5) with the IMPUTE package in R from Bioconductor. Finally, each metabolite was normalized (log transformed and then standardized to mean 0 and variance 1).

BPA and phthalates measures below the LOD were treated as LOD/sqrt(2). Since 4 of the phthalate mono-esters stem from the same parent compound, we calculated the molar sum of DEHP exposure (MECPP, MEHHP, MEHP, and MEOHP) at each trimester and ran models with that in addition to individual mono-esters. All BPA and phthalate data were adjusted for specific gravity, natural log-transformed, and standardized (mean 0, variance 1) prior to statistical analysis. There were missing data in the trimester-specific BPA and phthalate measures due to mothers missing one or more study visits. Thus, missing exposure data (n = 34, 34, and 10 for T1, T2, and T3, respectively) were imputed using K-nearest-neighbor imputation with K = 5, borrowing information from mother's age during pregnancy, years of education, marital status, cohort, and observed exposure values from available trimesters of pregnancy.

## Covariates

Height and weight were assessed as previously described by trained personnel at the study visit, and BMI were converted to age- and sex-specific z-scores according to WHO criteria [58]. Child's age was standardized to a similar scale (mean 0, standard deviation 1). Tanner staging (scale of 1 to 5) of participants was assessed by a trained physician at the study visit [59], and pubertal onset was defined as any pubic hair or genital/breast stage greater than one.

## Statistical analysis

Multivariable linear regression models were used to assess the associations between individual metabolites and individual trimester exposures, adjusting for children's age, BMI z-score, sex and pubertal onset. We expect both sex-dependent and–independent effects of phthalate exposure, and as such we also conducted sex-stratified analyses. Multiple comparisons were accounted for using the Benjamini-Hochberg procedure based on 572 models (for identified metabolites) and a significance threshold of 10% (q<0.1, where q stands for the adjusted p-value) [60]. Since BMI and puberty could be on the causal pathway (i.e. mediators) in the relationship between phthalates and metabolites, we also ran models excluding these variables and compared the results with those of the statistically significant associations from the main model. Effect estimates did not change substantially (<10%).

We performed additional analyses with phthalates that had the most significant associations (q<0.1) with metabolites, e.g., T3 MIBP among boys and T3 MECPP, MEHHP, and MCPP among girls. Since metabolites in similar biological pathways can be highly correlated with one another, we assessed the relationship between exposure-associated metabolites with each other. We ordered these metabolites by hierarchal clustering and produced a heatmap of their Pearson correlations. In this analysis, we included the metabolomics features significantly associated with exposure at q<0.1 and expanded the list to include features associated at an uncorrected p-value<0.05. The purpose of this analysis was to identify exposure-associated clusters of metabolites that were related to one another.

All statistical analyses were carried out using R (version > 3.2.0).

## Supporting information

**S1 Fig. Correlation among metabolites associated with maternal third trimester mehhp concentrations in peripubertal girls.** Metabolites were associated with T3 MEHHP exposure among girls at an uncorrected p-value<0.05 are included in the heatmap below; an asterisk * is next to the name of metabolites significant at q-value<0.1. The heatmap shows the Pearson correlation between these metabolites with each other, and metabolites are ordered by hierarchal clustering.
(TIF)

**S2 Fig. Clustering of metabolites associated with maternal third trimester mibp concentrations in boys.** Metabolites that were associated with T3 MiBP exposure among boys at an uncorrected p-value<0.05 are included in the heatmap below; an asterisk * is next to the name of metabolites significant at q-value<0.1. The heatmap shows the Pearson correlation between these metabolites with each other, and metabolites are ordered by hierarchal clustering.
(TIF)

**S1 Table. Trimester-specific maternal urinary phthalate concentrations before and after imputation.**
(DOCX)

**S2 Table. Number of associations with q<0.1 between all exposure measures and 572 metabolites among children.**
(DOCX)

## Acknowledgments

The authors acknowledge the research staff at participating hospitals and the American British Cowdray Hospital in Mexico City for providing research facilities. We thank the mothers and children for participating in the study.

## Author Contributions

**Conceptualization:** Jaclyn M. Goodrich, John D. Meeker, Martha M. Téllez-Rojo, Karen E. Peterson.

**Data curation:** Lu Tang, Jennifer L. Meijer, Wei Perng, John D. Meeker, Adriana Mercado-García, Alejandra Cantoral, Martha M. Téllez-Rojo, Karen E. Peterson.

**Formal analysis:** Jaclyn M. Goodrich, Lu Tang, Peter X. Song.

**Funding acquisition:** Jaclyn M. Goodrich, Wei Perng, Deborah J. Watkins, John D. Meeker, Martha M. Téllez-Rojo, Karen E. Peterson.

**Investigation:** Jaclyn M. Goodrich, Yanelli R. Carmona, Wei Perng, Deborah J. Watkins, Alejandra Cantoral, Karen E. Peterson.

**Methodology:** Jaclyn M. Goodrich, Jennifer L. Meijer, Deborah J. Watkins, Adriana Mercado-García, Peter X. Song, Martha M. Téllez-Rojo.

**Supervision:** Peter X. Song, Martha M. Téllez-Rojo, Karen E. Peterson.

**Visualization:** Yanelli R. Carmona, Jennifer L. Meijer.

**Writing – original draft:** Jaclyn M. Goodrich, Yanelli R. Carmona, Karen E. Peterson.

**Writing – review & editing:** Jaclyn M. Goodrich, Lu Tang, Yanelli R. Carmona, Jennifer L. Meijer, Wei Perng, Deborah J. Watkins, John D. Meeker, Adriana Mercado-García, Alejandra Cantoral, Peter X. Song, Martha M. Téllez-Rojo, Karen E. Peterson.

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
