## [Decision Letter · Decision Letter 0]

18 May 2022

PONE-D-21-23121Trimester-specific phthalate exposures in pregnancy are associated with circulating metabolites in childrenPLOS ONE

Dear Dr. Goodrich,

Thank you for submitting your manuscript to PLOS ONE. After careful consideration, we feel that it has merit but does not fully meet PLOS ONE’s publication criteria as it currently stands. Therefore, we invite you to submit a revised version of the manuscript that addresses the points raised during the review process.

Both reviewers recommend a careful review for concise text as well as improve clarity in addition tot he scientific questions.  Please carefully consider each comment and respond accordingly.  

We look forward to receiving your revised manuscript.

Kind regards,

Timothy J Garrett, PhD

Academic Editor

PLOS ONE

Journal Requirements:

3. Thank you for stating in your Funding Statement: "This study was funded by grants from the U.S. Environmental Protection Agency (US EPA, https://www.epa.gov/), grant numbers RD834800 (KEP) and RD83543601 (KEP), and from the National Institute for Environmental Health Sciences (NIEHS, https://www.niehs.nih.gov/) P20 ES018171 (KEP), P01 ES02284401 (KEP), R01 ES007821 (KEP), R01 ES01493 (MMTR), R01 ES013744 (MMTR), and P30 ES017885 (KEP, JMG). This study was also supported and partially funded by the National Institute of Public Health/Ministry of Health of Mexico (https://www.insp.mx/insp-overview.html; MMTR). The contents of this publication are solely the responsibility of the grantee and do not necessarily represent the official views of the funding agency. The funders had no role in study design, data collection and analysis, decision to publish, or preparation of the manuscript."

4. Thank you for stating the following in the Acknowledgments Section of your manuscript: "The authors acknowledge the research staff at participating hospitals and the American British Cowdray Hospital in Mexico City for providing research facilities. We thank the mothers and children for participating in the study. This study was made possible by U.S. Environmental Protection Agency (US EPA) grants RD834800 and RD83543601 and National Institute for Environmental Health Sciences (NIEHS) grants P20 ES018171, P01 ES02284401, R01 ES007821, R01 ES014930, R01 ES013744, and P30 ES017885. This study was also supported and partially funded by the National Institute of Public Health/Ministry of Health of Mexico. The contents of this publication are solely the responsibility of the grantee and do not necessarily represent the official views of the US EPA or the NIH.  Further, the US EPA does not endorse the purchase of any commercial products or services mentioned in the publication. "

Please remove any funding-related text from the manuscript and let us know how you would like to update your Funding Statement. Currently, your Funding Statement reads as follows: "This study was funded by grants from the U.S. Environmental Protection Agency (US EPA, https://www.epa.gov/), grant numbers RD834800 (KEP) and RD83543601 (KEP), and from the National Institute for Environmental Health Sciences (NIEHS, https://www.niehs.nih.gov/) P20 ES018171 (KEP), P01 ES02284401 (KEP), R01 ES007821 (KEP), R01 ES01493 (MMTR), R01 ES013744 (MMTR), and P30 ES017885 (KEP, JMG). This study was also supported and partially funded by the National Institute of Public Health/Ministry of Health of Mexico (https://www.insp.mx/insp-overview.html; MMTR). The contents of this publication are solely the responsibility of the grantee and do not necessarily represent the official views of the funding agency. The funders had no role in study design, data collection and analysis, decision to publish, or preparation of the manuscript. "

7. Please upload a new copy of Figures S1 and S2 as the detail is not clear. Please follow the link for more information: https://blogs.plos.org/plos/2019/06/looking-good-tips-for-creating-your-plos-figures-graphics/" https://blogs.plos.org/plos/2019/06/looking-good-tips-for-creating-your-plos-figures-graphics/

Reviewers' comments:

Reviewer's Responses to Questions

**Comments to the Author**

1. Is the manuscript technically sound, and do the data support the conclusions?

Reviewer #1: Partly

Reviewer #2: Partly

2. Has the statistical analysis been performed appropriately and rigorously? 

Reviewer #1: Yes

Reviewer #2: No

3. Have the authors made all data underlying the findings in their manuscript fully available?

Reviewer #1: Yes

Reviewer #2: Yes

4. Is the manuscript presented in an intelligible fashion and written in standard English?

Reviewer #1: Yes

Reviewer #2: Yes

5. Review Comments to the Author

Reviewer #1: Summary

The authors attempt to perturb biochemical pathways associated with exposure to phthalates during gestation, which influence the development of cardiometabolic risk? The goal is unclear, but I believe that the authors are looking to provide proof of concept that global metabolomics could identify new biomarkers in human subjects to elucidate the effects of exposure in later life, as mentioned later on in the manuscript.

Basic summary: Phthalates were measured during each trimester and correlated with the metabolic profiles, of the children, of the participants in later life (the author does a good job of justifying and identifying studies in animals that show similar findings to theirs).

However, overall, it is not clear what their goal(s) are- leaving many unanswered questions.

Importance

The paper lacks focus on the importance of the study. Pregnant women typically know that they should avoid smoking, alcohol, and raw fish, etc., but is everyone aware that they should probably avoid lipstick, perfumes, etc., probably not. However, if we were able to discover a biomarker related to the mechanism of phthalate exposure, how would we intervene given that the ‘damage’ has already been done? Or is the goal to search for biomarkers of cardiometabolic syndrome? I think the former, please clarify why such biomarkers are needed/how they will be helpful for clinical decision making/intervention.

Major Comments

1. Clarify the goal of the study- Is your goal to identify biomarkers for cardiometabolic exposure or to elucidate the mechanism of phthalate exposure to predict cardiometabolic disease? Initially I thought the former but as I read the manuscript the later seemed more fitting. What are the clinical markers of cardiometabolic risk?

2. Cardiometabolic disorders represent a cluster of interrelated risk factors, primarily hypertension, elevated fasting blood sugar, dyslipidemia, abdominal obesity, and elevated triglycerides. How will new biomarkers be useful/what aspect of the diagnostic decision making is hindered? Assuming we find biomarkers for associated with risk how will they help with intervention?

3. Did you have a control group? How do you know that the statistical differences were due to phthalate exposure during gestation, and not sex, age, diet, etc.? Admittedly challenging to control for in human studies!

4. Looks like a lot of negative ion FA’s being identified in the male (boy) cohort- have you looked at %CV variability of each given that smaller molecules do not always ionize well in negative mode?

5. What is the distribution of the female (girl) cohort in terms of age? Testosterone is noted as being statistically significant but may also be due to age. The clinical lab applies different age-based reference intervals for total- and free-testosterone. Might be useful to report tanner staging results by sex in the supplemental in addition to age distribution of females (girls).

6. How did you normalize the data and adjust for batch effects? More detail needed in the method section overall. What mass spectrometer was used- Thermo Tribrid IQX MS, for example. Liquid chromatography- mobile phase, gradient, column temperature, etc. What software was used for data analysis? Should break this section down into metabolomics and lipidomics as details will vary between the two.

7. What extraction was carried-out per-application, metabolomics vs lipidomics?

8. Not clear what the comparisons are. When you say ‘girls only comparison? Is this between trimesters? Clarify overall comparisons that were carried-out.

Reviewer #2: This manuscript reports associations between 9 maternal urinary phthalate metabolites and untargeted metabolomics profiles in serum samples collected from children aged 8-14 years. The results present a large number of phthalate associations across trimester with serum metabolites among the entire cohort of children as well as within gender/sex. The primary weakness of the paper include:

- the introduction could consolidated to focus on the underlying biology of maternal phthalate exposure and pediatric outcomes. As written, the introduction is very long and provides information that is not specific to the research objective. It seems as though some of this information could be moved to the discussion.

- methods are completely lacking information on how the metabolites were measured using mass spectrometry. Additionally, there is inadequate information on data processing software, code and techniques.

- it is unclear why the analysis was stratified by gender/sex rather than a formal interaction test.

- the discussion suffers from the same issue as the introduction. It is rambling and hard to discern what is important. I would recommend consolidating the discussion into discreet paragraphs related to: overall cohort results, sex specific results, rationale biological explanation that links underlying biology to the results.

- results/tables would benefit from including standardized metabolite ID's (HMDB, chemspider) that will help with replicating these results in future studies.

6. PLOS authors have the option to publish the peer review history of their article (what does this mean?). If published, this will include your full peer review and any attached files.

Reviewer #1: No

Reviewer #2: No

---

## [Author Response · Author response to Decision Letter 0]

8 Jul 2022

We have uploaded a letter responding to all reviewer and editorial comments.

---

## [Editor Report · Decision Letter 1]

27 Jul 2022

Trimester-specific phthalate exposures in pregnancy are associated with circulating metabolites in children

PONE-D-21-23121R1

Dear Dr. Goodrich,

We’re pleased to inform you that your manuscript has been judged scientifically suitable for publication and will be formally accepted for publication once it meets all outstanding technical requirements.

Kind regards,

Timothy J Garrett, PhD

Academic Editor

PLOS ONE
---

## [Editor Report · Acceptance letter]

19 Aug 2022

PONE-D-21-23121R1 

Trimester-specific phthalate exposures in pregnancy are associated with circulating metabolites in children 

Dear Dr. Goodrich:

I'm pleased to inform you that your manuscript has been deemed suitable for publication in PLOS ONE. Congratulations! Your manuscript is now with our production department. 

Kind regards, 

on behalf of

Dr. Timothy J Garrett 

Academic Editor

PLOS ONE